# Performance of a deep learning based neural network in the selection of human blastocysts for implantation

**Charles L Bormann[1,2†], Manoj Kumar Kanakasabapathy[3†], Prudhvi Thirumalaraju[3†], Raghav Gupta[3], Rohan Pooniwala[3], Hemanth Kandula[3], Eduardo Hariton[1], Irene Souter[1,2], Irene Dimitriadis[1,2], Leslie B Ramirez[4], Carol L Curchoe[5,6], Jason Swain[6], Lynn M Boehnlein[7], Hadi Shafiee[2,3]***

[1]Division of Reproductive Endocrinology and Infertility, Department of Obstetrics and Gynecology, Massachusetts General Hospital, Harvard Medical School, Boston, United States; [2]Harvard Medical School, Boston, United States; [3]Division of Engineering in Medicine, Department of Medicine, Brigham and Women's Hospital, Harvard Medical School, Boston, United States; [4]Extend Fertility, New York, United States; [5]San Diego Fertility Center, San Diego, United States; [6]Colorado Center for Reproductive Medicine IVF Laboratory Network, Englewood, United States; [7]Division of Reproductive Endocrinology and Infertility, Department of Obstetrics and Gynecology, University of Wisconsin, Madison, United States

**Abstract** Deep learning in in vitro fertilization is currently being evaluated in the development of assistive tools for the determination of transfer order and implantation potential using time-lapse data collected through expensive imaging hardware. Assistive tools and algorithms that can work with static images, however, can help in improving the access to care by enabling their use with images acquired from traditional microscopes that are available to virtually all fertility centers. Here, we evaluated the use of a deep convolutional neural network (CNN), trained using single timepoint images of embryos collected at 113 hr post-insemination, in embryo selection amongst 97 clinical patient cohorts (742 embryos) and observed an accuracy of 90% in choosing the highest quality embryo available. Furthermore, a CNN trained to assess an embryo's implantation potential directly using a set of 97 euploid embryos capable of implantation outperformed 15 trained embryologists (75.26% vs. 67.35%, p<0.0001) from five different fertility centers.

**\*For correspondence:**
HSHAFIEE@BWH.HARVARD.EDU

[†]These authors contributed equally to this work

## Introduction

Assisted reproductive technologies (ART) such as in vitro fertilization (IVF), while a solution to many infertile couples have been inefficient with an average success rate of approximately 30% reported in 2015 in the US (*CDC, 2015*). IVF is also an expensive solution costing patients well over $10,000 out-of-pocket for each ART cycle in the US with many patients requiring multiple cycles to achieve successful pregnancy (*CDC, 2015*; *Birenbaum-Carmeli, 2004*; *Toner, 2002*). Although multiple factors such as maternal age, medical diagnosis, gamete and embryo quality, and endometrium receptivity determine the success of ART cycles, the challenge of non-invasive selection of the highest available quality from a patient's cohort of embryos (top-quality embryo) for transfer remains as one of the most important factors in achieving successful ART outcomes (*Vaegter et al., 2017*; *Barash et al., 2017*; *Conaghan et al., 2013*; *Wong et al., 2013*; *Racowsky et al., 2015*; *Filho et al., 2010*; *Machtinger and Racowsky, 2013*; *Demko et al., 2016*; *Einarsson et al., 2017*; *Hill et al., 1989*; *Erenus et al., 1991*; *Paulson et al., 1990*; *Osman et al., 2015*).

**eLife digest** Around one in seven couples have trouble conceiving, which means there is a high demand for solutions such as in vitro fertilization, also known as IVF. This process involves fertilizing and developing embryos in the laboratory and then selecting a few to implant into the womb of the patient. IVF, however, only has a 30% success rate, is expensive and can be both mentally and physically taxing for patients. Selecting the right embryos to implant is therefore extremely important, as this increases the chance of success, minimizes complications and ensures the baby will be healthy.

Currently the tools available for making this decision are limited, highly subjective, time-consuming, and often extremely expensive. As a result, embryologists often rely on their experience and observational skills when choosing which embryos to implant, which can lead to a lot of variability. An automated system based on artificial intelligence (AI) could therefore improve IVF success rates by assisting embryologists with this decision and ensuring more consistent results. The AI system could learn how embryos develop over time and then uses this information to select the best embryos to implant from just a single image. This would offer a cheaper alternative to current analysis tools that are only available at the most expensive IVF clinics.

Now, Bormann, Kanakasabapathy, Thirumalaraj et al. have developed an AI system for IVF based on thousands of images of embryos. Using individual images, the system selected embryos of a comparable quality to those selected by a human specialist. It also showed a greater ability to identify embryos that will lead to successful implantation. Indeed, the software outperformed 15 embryologists from five different centers across the United States in detecting which embryos were most likely to implant out of a group of high-quality embryos with few visible differences.

Artificial intelligence has many potential applications to support expert clinical decision-making. Systems like these could improve success, reduce errors and lead to faster, cheaper and more accessible results. Beyond immediate IVF applications, this system could also be used in research and industry to help understand differences in embryo quality.

Traditional methods of embryo selection rely on visual embryo morphological assessment and are highly practice-dependent and subjective (*Storr et al., 2017*; *Baxter Bendus et al., 2006*; *Paternot et al., 2009*). Fully automated assessments of embryos are challenging owing to the complexity of embryo morphologies. Emulating the skill of highly trained embryologists in efficient embryo assessment in a fully automated system is a major challenge in all of the previous work done in computer-aided assessments of embryos due to focus on measuring specific expert-defined parameters such as zona pellucida thickness variation, number of blastomeres, degree of cell symmetry and cytoplasmic fragmentation, etc. (*Rocha et al., 2017a*; *Rocha et al., 2017b*).

Machine learning is loosely defined as a computer program that learns a given task over time through experience and improves itself to achieve the best possible task performance. In the past decade, advances in hardware compute performance and machine learning techniques have significantly improved their applicability in real-world medical and non-medical problems. Recently, machine learning has been proposed as a solution for automated analysis of embryo morphologies (*Rocha et al., 2017b*; *Bormann et al., 2020*; *Dimitriadis et al., 2019*; *Thirumalaraju et al., 2019*; *Khosravi et al., 2019*; *Kanakasabapathy et al., 2019a*). This work makes use of a deep convolutional neural network (CNN), a representation learning technique, that has been proven to be effective in image classification tasks. Unlike most prior computer-aided algorithms, including some techniques of machine learning used for embryo assessment, the reported CNN architecture allows automated embryo feature selection and analysis at the pixel level without any interference by an embryologist (*Rocha et al., 2017a*; *Rocha et al., 2017b*). Such networks do not depend on human-specified features and can develop an ability to evaluate embryos categorically through iterative learning from thousands of examples. The use of deep-learning in IVF has also been explored; however, these recent neural network-based approaches have focused on either classifying embryos based on morphological quality and were not evaluated for transfer outcomes, or were developed with the use of time-lapse series of images toward the evaluation of implantation (*Khosravi et al., 2019*; *Tran et al., 2019*). It is important to emphasize here that most fertility centers do not possess

time-lapse imaging hardware even in the United States of America (*Dolinko et al., 2017*). The lack of availability of such hardware limits an otherwise promising technology mostly to resource-rich settings and fail to improve quality of and access to care in resource-constrained settings where such advances are sorely needed (*Wahl et al., 2018*; *Hosny and Aerts, 2019*). Furthermore, in current clinical practice, embryos with the highest morphological grades (top-quality) are the first to be transferred, however, clinically these decisions are performed manually, even with time-lapse imaging systems. The development of networks that can measure an embryo's potential for implantation and help in rank ordering embryos in a patient embryo cohort for transfer have utility in virtually all fertility centers.

Conventionally, embryo transfers are performed at the cleavage or the blastocyst stage of development. Embryos are at the cleavage stage 2–3 days after fertilization and develop further in suitable culture conditions to reach the blastocyst stage 5–7 days after fertilization. Blastocyst embryo transfers, in particular, have been associated with better implantation rates and have helped lower the number of embryos transferred at a time (*De Croo et al., 2019*). Therefore, in this study, we have investigated the use of a CNN pre-trained with 1.4 million ImageNet images and transfer-learned using 2440 static human embryo images recorded at a single time-point of 113 hr post insemination (hpi) for the development of neural networks that can help identify embryos capable of implantation and for identifying the top quality embryos (*Figure 1*). The top-quality embryos were identified by combining a previously developed network (Xception architecture) trained to classify embryos based on its blastocyst quality with a genetic algorithm scheme (*Figure 1*; *Thirumalaraju et al., 2020*). The original neural network was trained on a hierarchical system of categorization, derived from a clinical Gardener-based grading system, to minimize data sparsity and improve overall network learning (*Kanakasabapathy et al., 2019a*; *Thirumalaraju et al., 2020*; *Kanakasabapathy et al., 2019b*; *Esteva et al., 2017*). The two major categories of non-blastocysts and blastocysts made up the inference classes, which included the training classes 1, 2, and 3, 4, 5, respectively (*Figure 1*). Pre-training with a large dataset of images from ImageNet honed the ability of the developed CNN to identify the shape, structure, and texture variations between morphologically complex embryos with minimal data requirements while the genetic algorithm helped in rank ordering embryos by generating unified scores (*Figure 1*). The developed network was evaluated using an independent test set comprising of 97 patient-embryo cohorts. Embryos of the highest quality that were selected from the patient cohorts were evaluated using known implantation outcomes.

Additionally, we also investigated if the neural network can be trained to directly differentiate between embryos based on their potential for implantation (*Figure 1*). Our tests with patient cohorts using the algorithm does not account for the ploidy status of the embryos. Since pre-implantation genetic screened (PGS) euploid embryos are associated with higher implantation chance, we also designed a neural network to evaluate the network performance in refining the screened embryos based on their implantation potential. The evaluations using the patient cohorts tend to yield embryo selections with unknown outcomes or ploidy status, therefore, for this section of the study, we utilized a test set of 97 euploid embryos with known implantation outcomes. The CNN was trained and evaluated in identifying euploid embryos capable of implantation and the performance was compared against those of 15 embryologists from five different fertility centers across the United States of America.

## Results

### Evaluation of embryo selection based on embryo quality

In our evaluations of the CNN in categorizing embryos imaged at 113 hpi based on their morphology, the network performed with an accuracy of 90.97% (area under the curve: 0.96) in differentiating between blastocysts and non-blastocysts (n = 742) (*Kanakasabapathy et al., 2019a*; *Thirumalaraju et al., 2020*; *Figure 2—figure supplement 1*). The high accuracy indicated that the trained network was concordant with embryologists in categorizing embryos. These categorization scores (five values per embryo) need to be used by taking into account the scores of other embryos in the cohort to establish a rank order. In order to use the five probability values effectively for calculating the embryo score, we utilized a genetic algorithm, which is well-suited for optimization

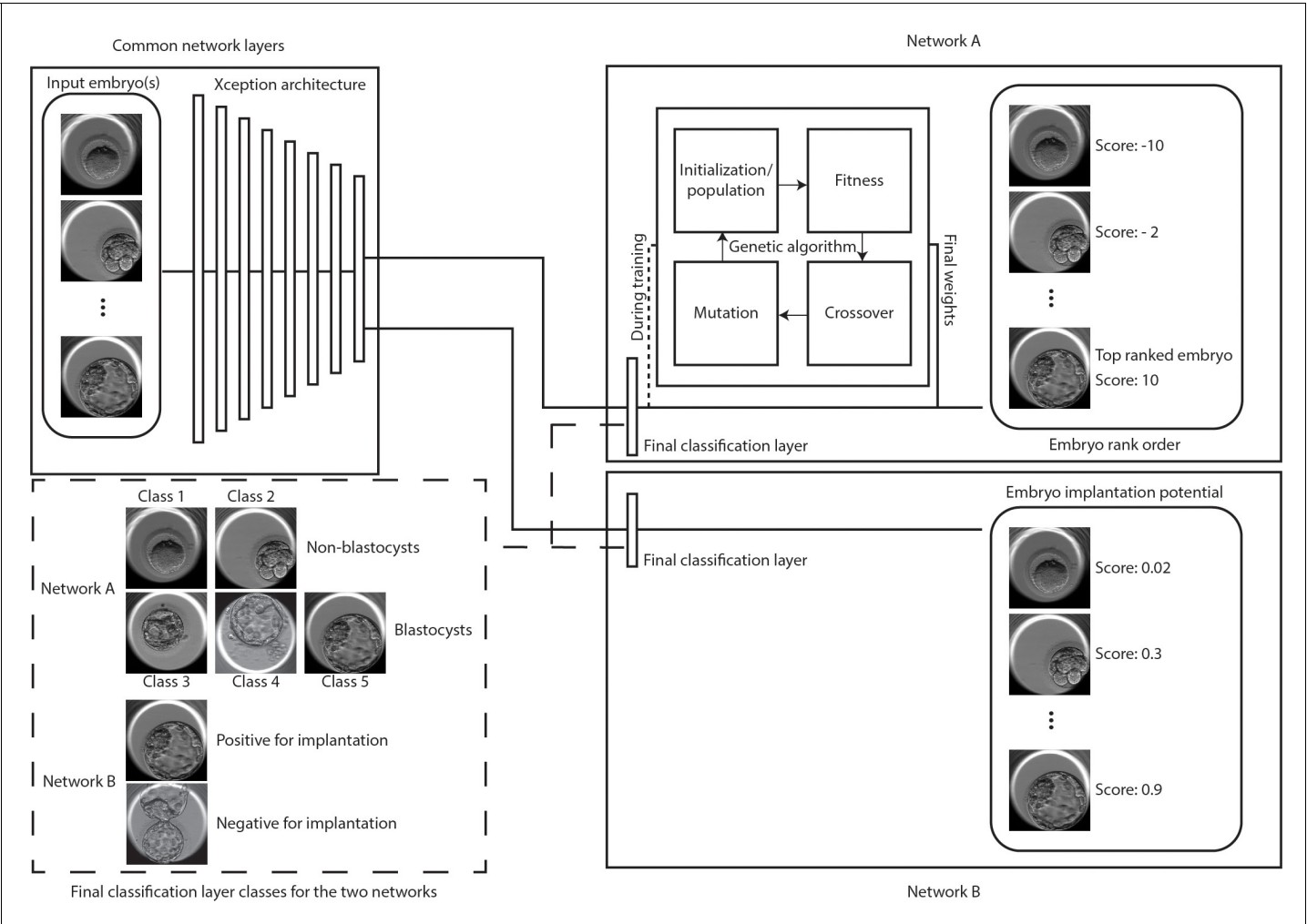

**Figure 1.** Classification and selection of embryos at 113 hpi. The schematic shows neural networks that classify, and rank order embryos based on their morphological quality (network A) and classify embryos based on the implantation potential (network B). The two networks share a common Xception architecture but the classification layers are specific to each task. Network A also uses a genetic algorithm that helps in generating embryo scores by using the softmax output of the network with weights generated by the algorithm during training. Embryo(s) with the highest scores are evaluated for single embryo and double embryo transfer scenarios using the retrospective test set. The implantation potential is given by the softmax output of the neural network.

problems with multiple existing solutions. Here, the genetic algorithm empowered the developed CNN to make selections of the top-quality embryos within a patient's embryo cohort at 113 hpi. Therefore, once we established that the network was capable of categorizing embryos based on their morphologies with high accuracy, we used a genetic algorithm and the network defined probability values of the embryos, belonging to each of the five training classes, to rank order the embryos for transfer. The $5 \times 1$ vector weights generated by the genetic algorithm during its training phase were used in evaluating retrospectively collected embryo cohorts from 97 patients. The final weights utilized in this study were $-10.01226347$, $-3.63697951$, $-3.32090987$, $2.15367795$, and $2.8715555$ for classes 1 through 5, respectively. Embryos were ranked by the algorithm from highest to the lowest.

According to the American Society for Reproductive Medicine guidelines on the limits to the number of embryos per transfer, one embryo is transferred for high prognosis patients with <37 years of age and two or more embryos are transferred for patients with >37 years of age as well as younger patients with low prognosis (***Practice Committee of the American Society for Reproductive Medicine. Electronic address: ASRM@asrm.org and Practice Committee of the Society for Assisted Reproductive Technology, 2017***). Therefore, in this study, the selection accuracy was

assessed for scenarios of single embryo transfers (SET) and double embryo transfers (DET). Using embryo cohort images (n = 732) from the 97 patients, the accuracy of 5 well-trained embryologists' selections were evaluated in comparison to selections made by the CNN + genetic algorithm (CNNg). The rank-ordering performed by the algorithm may not utilize the same features used by embryologists in identifying the top embryos for transfer. Therefore, we initially evaluated the ability of both groups to effectively select (i) blastocyst(s) for transfer and (ii) the highest quality of blastocyst(s) (HQB) available for transfer. High-quality blastocysts are defined as embryos that met the freezing criteria (>3 CC blastocyst grade; see Materials and methods) of the Massachusetts General Hospital (MGH) fertility clinic.

For blastocyst selections at 113 hpi, the CNNg algorithm performed with an accuracy of 98.96% for SET, which was similar (p>0.05) to the average accuracy of the embryologists (96.91%, CI: 94.69% to 99.12%) (n = 5) (*Figure 2A*). However, when two embryo selections for DET were allowed based on blastocyst and non-blastocyst classification, the CNNg algorithm performed with an accuracy of 100.00%, which was better (p<0.05) than embryologists (n = 5) who performed with an average accuracy of 98.76% (CI: 97.69% to 99.83%) (*Figure 2B*).

Toward the selection of HQB at 113 hpi, the accuracy of the CNNg algorithm for SET was 89.69% similar (p>0.05) to the embryologists (n = 5) who performed with an average accuracy of 90.31% (CI: 87.50% to 93.11%) (*Figure 2C*). When two embryo selections for DET at 113 hpi were allowed, the system performed with a better (p<0.05) accuracy of 97.94% in comparison to the embryologists who performed with an average accuracy of 96.91% (CI: 96.00% to 97.81%) (*Figure 2D*). The evaluations indicated that the two groups made selections that were of similar quality or marginally different quality. Since the network was trained on the MGH classification criteria, the comparable performance of the CNNg algorithm and embryologists indicated that the neural network has trained itself sufficiently and made selections that were of clinically acceptable quality. In our evaluations, the selections made by each group, while were of similar quality, were observed to not necessarily be the same embryos from each cohort, and thus their transfer outcomes may be different.

## Evaluation of selections using implantation outcomes

It is critical to evaluate the system performance in selecting the patient embryos based on pregnancy (implantation) outcome. Typically, in a clinical IVF cycle, the top-quality embryo is selected from the cohort of available embryos and is transferred to the patient. Embryos, which are similarly of a high-quality, are often frozen based on the freezing criteria used by the fertility center, for transfers in subsequent procedures for the same patient if needed. Frozen cycle transfers are not performed for all patients. Hence, the CNNg algorithm was evaluated in embryo selection for SET at 113 hpi using patient embryo cohorts based on actual implantation outcomes of the selected embryos and

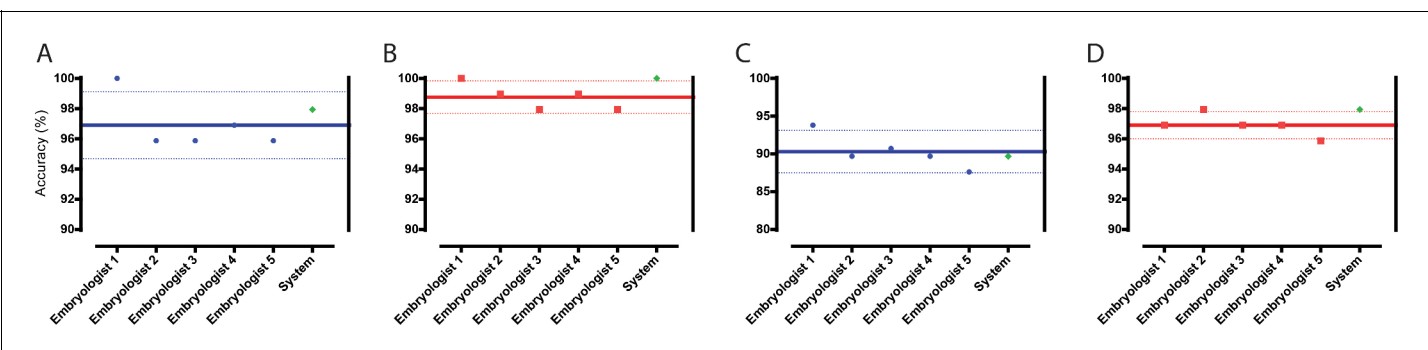

**Figure 2.** Classification and selection of embryos at 113 hpi. (**A**) The performance in single embryo selections by embryologists and the algorithm in selecting blastocysts using embryo morphologies obtained at 113 hpi from 97 patient cohorts. (**B**) The performance in double embryo selections by the two groups in selecting blastocysts (n = 97 patient cohorts). (**C**) The performance in single embryo selections by the two groups in selecting the highest quality blastocysts (n = 97 patient cohorts). (**D**) The performance of the two groups in selecting the highest quality blastocysts when two selections were provided (n = 97 patient cohorts).

The online version of this article includes the following figure supplement(s) for figure 2:

**Figure supplement 1.** Confusion matrix of the network in classifying embryos based on their morphological quality.

associated cycle characteristics (n = 97) are provided in *Supplementary file 1*-table 1. The test dataset was retrospectively collected based on pre-defined selection criteria and evaluations of transfer outcomes were performed using fresh embryo transfer cycles. The system selected 97 embryos in 97 patient embryo cohorts (742 embryos in total), out of which 44 embryos had known implantation outcomes. The accuracy of the system in SET through embryo selection at 113 hpi based on its implantation outcome was 59.1% while the implantation success rate for the 102 transferred embryos at the MGH fertility center was 44.1% for blastocyst transfers (*Supplementary file 1*-table 2). Furthermore, prior reports suggest that in general practice, the average implantation rates for manual-based embryo selection and transfers at blastocyst stages can be as low as 34% (*Martins et al., 2017*).

A limitation of a retrospective study is that not all embryos are transferred. Implantation outcomes of all embryos selected by the CNNg algorithm cannot be evaluated. Therefore, although the dataset was prepared not taking into consideration the availability of subsequent frozen cycle transfers, we investigated with the fertility center if the patients of the test set had any subsequent embryo transfers using the frozen embryos from the test set. When we consider subsequent frozen embryo transfers, five embryos originally selected by the CNNg algorithm at 113 hpi had known implantation outcomes of which four led to successful implantations (*Supplementary file 1*-table 2). The accuracy of the CNNg algorithm in SET, when both fresh and frozen embryo transfers were considered, was 61.2%. In such a scenario, for this specific dataset, the implantation success rate at MGH fertility center was 48.5% for blastocyst transfers when including both frozen and fresh transfers. The results suggest that the CNNg algorithm has the potential to improve clinical transfer outcomes. It should, however, be emphasized that in this particular analysis the performance of the system was evaluated by only using the embryos selected by the network and the embryologists.

Furthermore, to evaluate if a CNN can potentially measure implantation potential through morphology alone, a pooled set of 29 embryo images with known transfer outcomes in a pilot study was used by the network to evaluate embryos based on their potential for implantation. The network was trained as a binary classifier and the SoftMax probability values outputted by the network was used as the embryo's implantation potential. The CNN was retrained using 281 embryo images with known implantation outcomes that did not overlap with the test set and the final classification layer was replaced with the two classes- negative implantation and positive for implantation. The ability to differentiate embryo was measured through a receiver operating characteristic curve (ROC) analysis, establishing area under the curve (AUC) of 0.771 (CI: 0.579 to 0.906) ($p<0.05$) and the CNN performed with an accuracy of 82.76% (CI: 64.23% to 94.15%) (*Figure 3A*). Ten out of 11 embryos had implanted with an implantation potential of over 0.47 and similarly, for embryos that scored less than 0.47, 12 out of 18 embryos did not implant according to the patient cycle history.

## Evaluation of euploid embryos based on their implantation potential

After we observed high performance in the artificial intelligence (AI)-based implantation potential prediction when compared with historical clinical data, we further conducted a multi-center AI system evaluation by comparing the implantation potential prediction accuracies obtained from the AI system and the embryo selections of 15 embryologists from five different fertility clinics. Here, we used 97 genetically screened euploid embryos transferred at 113 hpi to remove the effect of chromosomal abnormalities as a confounder, which existed in the pilot study (29 patient embryo). The IVF cycle characteristics in which these embryos were used are provided in *Supplementary file 1*-table 3. The system performed with an accuracy of 75.25% while the embryologists performed with an average accuracy of 67.35% (CI: 64.52% to 70.19%) in differentiating euploid embryos based on their implantation outcome (*Figure 3B*). A one-sample t-test revealed that the CNN significantly outperformed ($p<0.05$) the embryologists in predicting embryo implantation by measuring the implantation potential of euploid embryos using a static image obtained at a single time-point of 113 hpi. The average implantation score of euploid embryos misclassified based on their implantation outcome using the CNN was 0.57. 95% of the misclassified euploid embryos possessed scores ranging between 0.51 and 0.63. Implantation scores closer to 0.5 indicate lower confidence in system predictions while implantation scores closer to 0 or 1 indicate higher confidence in system predictions (*Figure 3—figure supplement 1*). These results indicate that the majority of system errors in misclassifying the euploids occur among the embryos with the lowest confidence. Approximately 91% of euploid embryos with implantation potential scores of 0.80 or higher, and nearly 81% of

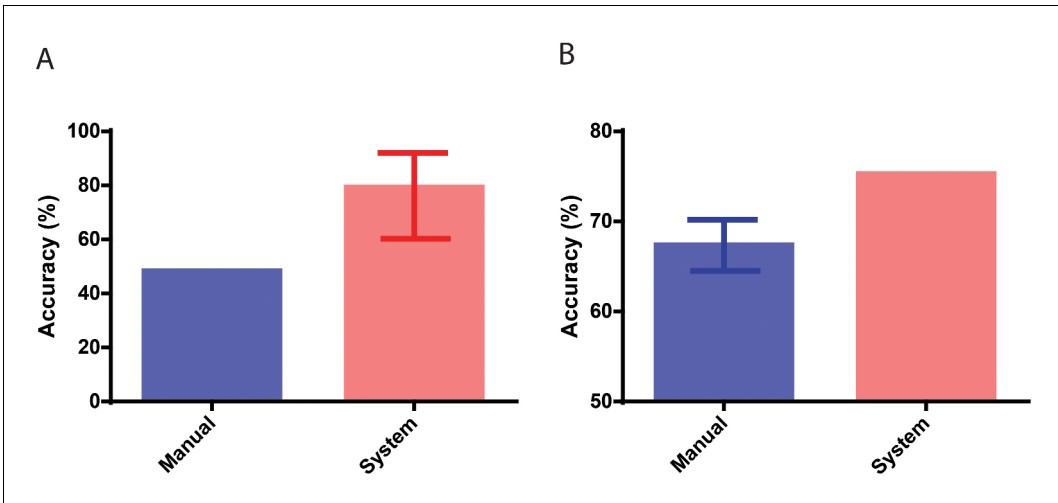

**Figure 3.** Performance in identifying embryos based on implantation outcomes. (**A**) The performance of the neural network system in identifying embryos that implanted compared to the baseline historical implantation for the image set (n = 29). The error-bar represents the Clopper-Pearson exact binomial 95% confidence interval. (**B**) The performance of the neural network system in identifying euploid embryos that implanted compared to the performance of 15 embryologists in identifying implanting embryos (n = 97). The error-bar represents the 95% confidence interval of the embryologists' performance in identifying implanting embryos.

The online version of this article includes the following figure supplement(s) for figure 3:

**Figure supplement 1.** Implantation potential and the relative implantation rates using the euploid embryo test set.

embryos with implantation potential scores above 0.66 successfully implanted when transferred (*Figure 3—figure supplement 1*). Similarly, around 78% of euploid embryos with an implantation potential <0.33, failed to successfully implant when transferred (*Figure 3—figure supplement 1*). These results suggest that the network's implantation scores agree well with transfer outcomes even in high-quality euploid embryos.

## Discussion

Deep neural networks hold value in aiding clinical decision making and have received significant attention from the IVF community. The deep-neural network-based approach showcased here is an objective approach to one of the more subjective but important parts of a clinical IVF process-embryo selections for transfer (*Bormann et al., 2020*). Since over 80% of fertility clinics rely on non-time lapse imaging systems as part of their clinical processes, such neural network-based algorithms that rely purely on static single timepoint images can effectively assist in decision making (*Dolinko et al., 2017*). In our study, we have evaluated two neural network-based approaches for improving embryo selection.

Firstly, we have demonstrated that a deep-neural network in combination with a genetic algorithm (CNNg) can yield a continuous score that represents the quality of the embryo and that objective orders of transfer can be determined for a given set of embryos using such scores. The ranking algorithm studied here was able to consistently select embryos of the highest available morphological quality. Although the network was trained to classify embryos based on their quality, it performed well even in differentiating between embryos of the same class when combined with a genetic algorithm. The benefit of such systems is particularly evident in cases where selections made by the clinic/embryologist, although of similar grade, resulted in lower overall transfer success rates. Our networks only focused on the morphological features for embryo quality assessments due to data scarcity. The network's learning can be compounded with data from additional timepoints, morphokinetics, and patient and cycle-specific information for more personalized IVF predictions and outcomes. Recently, Tran et al. studied the use of a deep-learning model (IVY) that can analyze

whole time-lapse videos instead of specific time points for fetal heartbeat prediction (*Tran et al., 2019*). However, the study was flawed since embryos with unknown outcomes (non-transferred embryos) were considered as negative outcome cases, which made up most of their dataset (~90%). The heavy class bias in their dataset and improper study design severely limits any conclusions that can be drawn from the work. A major hurdle for the development of networks capable of analyzing multi-timepoint images and with additional patient-specific information is the limited availability of diversified data with known clinical outcomes. During training, the lack of availability of such data prevents the networks from effectively learning relevant outcome-associated patterns in data. The need for data scales with the complexity of the task and the number of variables introduced. While this work focuses primarily on the utility of deep-learning algorithm for embryo evaluations at 113 hpi, it is also possible to develop similar networks for embryo evaluations at different timepoints, provided that sufficient data with matched outcomes/annotations are available. We have evaluated a similar network for use with cleavage-stage embryos (70 hpi) and showed that deep-learning approaches can outperform trained embryologists in certain tasks such as embryo selection (*Thirumalaraju et al., 2019*; *Kanakasabapathy et al., 2020*).

A major concern in any clinical practice, however, is the loss of viable embryos due to system errors. Therefore, the AI-based embryo selection algorithm reported here does not make any suggestion on discarding embryos. All embryos assessed by the CNNg in the selection process may be cryopreserved as per clinical practice. Thereby, our approach will not negatively affect the cumulative pregnancy rate since viable embryos will not be lost. However, it may improve the pregnancy rate as the system may be able to improve the chance of achieving a pregnancy faster with fewer embryos transferred. Furthermore, it is important to note that in its current stage this system is intended to act only as an assistive tool for embryologists. The embryologists can include the system's prediction to make better judgments during embryo selection. The scores provided by the algorithm are continuous, but it can also be easily modified to present its scoring results in both binary and a more categorical format.

Clinically, besides morphological features, various other important metrics and parameters are considered by embryologists at the time of decision making such as taking into account the ploidy status of the transferable embryos. PGS verified euploid embryos have been shown to possess a higher probability of successful outcome but cost a hefty premium on top of the cycle costs at most fertility centers in the United States (*Drazba et al., 2014*). Furthermore, for patients with two or more euploid embryos, additional assessments of embryo morphology are required to select the best embryo based on their morphology for transfer, since euploids do not inherently guarantee implantation. Thus far, to the best of our knowledge, no system, deep-learning-based or otherwise, has been shown to be capable of differentiating between euploid blastocysts based on their capacity for implantation. Euploid embryos are usually of the highest available quality and differentiating between them objectively and reliably through manual analysis can be extremely challenging. The CNN-based approach, through direct estimations of implantation potential from 113 hpi embryo morphology, outperformed trained embryologists in identifying implanting embryos from a set of PGS euploid embryos. This accomplishment exhibits the potential of artificial intelligence-based approaches to improve success rates in the IVF lab. Our observations indicated that the system performed with a significantly better agreement with the actual implantation outcome for embryos with implantation scores closer to 1 or 0 (Higher confidence). Furthermore, the comparison between the decisions made by 15 embryologists from different fertility centers in the US and the deep-neural network showcased that neural networks can outperform embryologists in identifying embryos capable of implantation. Hence, by applying the suggestions of a CNN, a trained embryologist can improve their selection of the embryo with the highest implantation potential.

Advances in artificial intelligence have fostered numerous applications that have the potential to improve standard-of-care in the different fields of medicine. While other groups have also evaluated different use cases for machine learning in assisted reproductive medicine, this approach is novel in how it used a CNN trained on a large dataset to make predictions based on static images. The approach has shown the potential of CNNs to be used in aiding embryologists to select the embryo with the highest implantation potential, especially amongst high-quality euploid embryos. Although the current retrospective study shows that these systems can perform better than highly-trained embryologists, randomized control trials are required before routine use in clinical practice is adopted.

## Materials and methods

### Data collection and preparation

Data were collected at the Massachusetts General Hospital (MGH) fertility center in Boston, Massachusetts. We used 3469 recorded videos of embryos collected from 543 patients with informed consent for research and publication, under an institutional review board approval for secondary research use. Videos were collected for research after institutional review board approval by the Massachusetts General Hospital Institutional Review Board (IRB#2017P001339 and IRB#2019P002392). All the experiments were performed in compliance with the relevant laws and institutional guidelines of the Massachusetts General Hospital, Brigham and Women's Hospital, and Partners Healthcare. The videos were collected using a commercial time-lapse imaging system (Vitrolife Embryoscope). The imaging system used a Leica 20x objective that collected images at 10-min intervals under illumination from a single 635 nm LED. Each patient's set of embryos were exported as videos (.avi) using the imaging system software. The videos of individual embryos were broken down into their respective frames to extract images from all timepoints post insemination. The images were identified by their timestamps and only images collected at 113 ± 0.05 hr post insemination were processed and used in this study. The extracted images were 250 × 250 pixels and they were cropped to 210 × 210 pixels. The cropping removed both the timestamps and identifiers present in the frame. All embryos used in the study were annotated using images from the fixed timepoints (113 hpi) by senior-level embryologists with a minimum of 5 years of human IVF training. Annotations for embryo implantation were assigned based on clinical outcomes. Out-of-focus images were included in the datasets and used for both testing and training. Only images of embryos that were completely non-discernable were removed from the study as part of the data cleaning procedure.

### Hierarchical categorization

The two networks in this study used two categorization systems. The network focused on the rank ordering of embryos used a hierarchical categorization system. The embryo images at 113 hpi time point were categorized between training classes 1 through five as described in detail elsewhere (*Thirumalaraju et al., 2020*). Briefly, degenerated embryos, which did not begin compaction formed Class 1 while Class 2 embryos were those that reached the morula stage by 113 hpi. Classes 1 and 2 together formed 'non-blastocysts' inference class. Class three embryos exhibit features of an early blastocyst which is highlighted by the presence of blastocoel cavity and thick zona pellucida but lack expansion. Class four embryos were blastocysts with blastocoel cavities occupying over half of the embryo volume but either their inner cell mass (ICM) or trophectoderm (TE) was of poor quality. They are non-freezable quality embryos (<3 CC), where three represents the degree of expansion (range 1–6) and C represents the quality of ICM and TE (range A-D), respectively. Class 5 embryos, however, met cryopreservation criteria (>3 CC) and included full blastocysts to hatched blastocysts. Classes 3, 4, and 5 together formed 'blastocysts' inference class. The two inference classes are used since the differentiation of blastocysts and non-blastocysts is a universally accepted categorization that is relevant to embryologists, while the five class categorization is specific to the neural network training, performance and evaluation (*Thirumalaraju et al., 2020*). Networks that were focused on estimating an embryo's implantation potential used a two-class training and inference system- positive for implantation and negative for implantation.

### Neural network training for 113 hpi

The 113 hpi evaluation dataset included images of 2440 embryos categorized across five classes post-cleaning based on their clinical annotations made at 113 hpi. Our training set for this classification task used 1188 images with a validation dataset of 510 images obtained at 113 hpi. With the availability of unskewed validation sets prior to augmentation, we used a data generator during training, which performed random rotations and flips across all classes on the fly. The system performing with an accuracy of 90.97% was used in this study in combination with our genetic algorithm. The genetic algorithm was trained and tested with the training data prior to testing it with our independent test data. No human interaction was required/performed once the images were provided to the system during testing, as the entire process was fully automated. The independent non-

overlapping test set consisted of 742 images of embryos originating from 97 patients. The selections were compared with embryologist selections. The network was also trained to classify embryos with successful and unsuccessful implantation. 281 embryo images with known implantation outcomes were used for training. Implantation signifies the attachment of a blastocyst into the endometrium. The status of implantation was clinically verified by ultrasound ~6 weeks after embryo transfer. Ninety-seven euploid embryos were evaluated by 15 embryologists, including director level embryologists from five different fertility centers.

### Embryo selection algorithm development

A genetic algorithm was designed to perform selections in combination with the neural network. The genetic algorithm component utilizes the probability scores of every embryo belonging to each of the five different classes to generate a transfer score that can be used to effectively identify the best embryo available in a cohort. For system evaluations, we used an independent set of embryos (100 patients; 2–12 embryos per patient), with no overlap with the training data set used for any prior exercise. The patient cohorts were chosen under the following criteria: (i) each patient embryo cohort had to possess at least two 2PN embryos, and (ii) at least one embryo of the patient embryo cohort developed to blastocyst stage by 113 hpi.

### Genetic algorithm

We trained a genetic algorithm to select the morphologically highest quality embryo from a given cohort. There are four phases namely initialization, selection, crossover, and mutation. The classified embryos for each patient were sorted according to their identifier numbers allotted by the deep neural network. A population of weights was generated at random during initialization. A population size of 100 was generated with a $5 \times 1$ matrix representing each weight. Each weight defined a possible solution for the rank-ordering of embryos based on their quality using the five training classes. The dot product of the weights with the output logits provided by the CNN was used in the calculation of the fitness. The algorithm runs multiple cycles to select the optimal set of weight towards achieving the appropriately suitable rank order of embryos based on their qualities. At each cycle, all the weight sets obtained using the given population were used rank-ordering embryos within the training set. The best 20 wt sets were selected in each cycle.

These selected weights (specimens) were then bred with each other with a probability set to 20%. It randomly selected two specimens from the selected top pool and created a random binary $5 \times 1$ matrix, where one represents that the given element should be switched in cohort and 0 represents that given element should not be switched within the cohort. The fitness function checks if the selected embryo belongs to the highest class available within the tested cohort. It checks if the selected solution (specimen) picked the embryo belonging to the top class in a given cohort of patient embryos. If the selected embryo belonged to the top class, the score was increased and if it did not, the score was not modified. After iterating for all patients' cohorts, the total scores were used to select the best 20 weights of the given population and were taken for crossover and mutation to repeat the process. The new specimens replaced their parents in the top selected group of embryos. Otherwise, the matrix remained the same. After breeding, each specimen from the top selected group was mutated to give five mutations by adding a random float $5 \times 1$ matrix with a probability of 20%. These mutations were then added to the new population and the selection step was repeated with the new population of 100. The genetic algorithm ran until the entire population converged to the same score after which a random weight was selected from the population as the final weight. Thus, final generated weights were used to further test the embryo cohorts within our test set.

## Acknowledgements

The authors thank embryology staff from Massachusetts General Hospital for participating in this study. The authors thank the Massachusetts General Hospital and Brigham and Women's Hospital Center for Clinical Data Science for their support and fruitful contributions.

## Additional information

### Competing interests

Charles L Bormann: Charles L Bormann, Ph.D has a patent WO2019068073A1 pending. Manoj Kumar Kanakasabapathy: Manoj Kumar Kanakasabapathy has a patent WO2019068073A1 pending. Prudhvi Thirumalaraju: Prudhvi Thirumalaraju has a patent WO2019068073A1 pending. Hadi Shafiee: Dr. Shafiee has a patent WO2019068073A1 pending. The other authors declare that no competing interests exist.

### Funding

| Funder | Grant reference number | Author |
|---|---|---|
| National Institutes of Health | R01AI138800 | Hadi Shafiee |
| Brigham and Women's Hospital | Precision Medicine Program | Hadi Shafiee |
| Partners Healthcare | Innovation Discovery Grant | Charles L Bormann Hadi Shafiee |
| National Institutes of Health | R61AI140489 | Hadi Shafiee |

The funders had no role in study design, data collection and interpretation, or the decision to submit the work for publication.

### Author contributions

Charles L Bormann, Manoj Kumar Kanakasabapathy, Conceptualization, Investigation, Data curation, Funding Acquisition, Project administration, Supervision, Validation, Formal analysis, Supervision, Methodology, Writing - original draft; Prudhvi Thirumalaraju, Conceptualization, Investigation, Validation, Software, Formal analysis, Methodology; Raghav Gupta, Rohan Pooniwala, Data curation, Software, Methodology, Formal analysis; Hemanth Kandula, Data curation, Software, Methodology, Validation, Formal analysis; Eduardo Hariton, Data curation, Writing - review and editing; Irene Souter, Resources, Data curation; Irene Dimitriadis, Resources, Data curation, Writing - review and editing; Leslie B Ramirez, Carol L Curchoe, Jason Swain, Lynn M Boehnlein, Resources, Supervision, Writing - review and editing; Hadi Shafiee, Conceptualization, Resources, Supervision, Project Administration, Funding acquisition, Methodology, Writing - review and editing

### Author ORCIDs

Manoj Kumar Kanakasabapathy https://orcid.org/0000-0002-0756-8585
Prudhvi Thirumalaraju https://orcid.org/0000-0002-0573-0243
Hadi Shafiee https://orcid.org/0000-0003-2240-7648

### Ethics

Human subjects: Embryo image/video data collected from patients were used in this study with an institutional review board approval (IRB#2017P001339) We used 3,469 recorded videos of embryos collected from patients with informed consent for research and publication, under an institutional review board approval for secondary research use.

### Decision letter and Author response

Decision letter https://doi.org/10.7554/eLife.55301.sa1
Author response https://doi.org/10.7554/eLife.55301.sa2

## Additional files

### Supplementary files

• Supplementary file 1. Patient and Cycle characteristics. (A) Patient population characteristics. All embryo images (except the PGT screened embryos) utilized for experiments reported in the study

were obtained from cycles that belong to the presented distribution of parameters. All values in table are presented as median along with the range unless noted otherwise. (**B**) Total number of transfer outcomes for embryos selected by the network. A total of 102 fresh-transfer embryos had known implantation outcomes (45 embryos implanted). Twenty-eight frozen transfers were performed by the clinic where 18 implanted. The table lists only embryos which were selected by the network with known outcomes for both fresh cycles and in frozen subsequent transfers. (**C**) Cycle characteristics of the euploid test set. Embryos used in the euploid embryo differentiation experiment based on the implantation outcomes, originated from cycles that belong to presented distribution of characteristics. These cycles are independent of the original 97 patient cohort test set and also the training data sets. All values in table are presented as median along with the range unless noted otherwise.

• Transparent reporting form

### Data availability

Patients did not explicitly consent to their data being made public and access is therefore restricted. Requests for the anonymized data should be made to Charles Bormann (cbormann@partners.org) and Hadi Shafiee (hshafiee@bwh.harvard.edu). Requests will be reviewed by a data access committee, taking into account the research proposal and intended use of the data. Requestors are required to sign a data-sharing agreement to ensure patients' confidentiality is maintained prior to the release of any data.

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
