## [Decision Letter]

**Acceptance summary:**

This paper is successful example of an emerging trend in medicine in which machine learning algorithms are employed to integrate clinical data in a rigorous and consistent manner in ways that outperform trained medical specialists in making critical decisions – in this case selecting embryos for implantation following in-vitro fertilization.

**Decision letter after peer review:**

Thank you for submitting your article "Performance of a deep-learning based neural network in selection of human blastocysts for implantation" for consideration by *eLife*. Your article has been reviewed by three peer reviewers, and the evaluation has been overseen by a Reviewing Editor and Michael Eisen as the Senior and Reviewing Editor. The reviewers have opted to remain anonymous.

The reviewers have discussed the reviews with one another and the Reviewing Editor has drafted this decision to help you prepare a revised submission.

As the editors have judged that your manuscript is of interest, but as described below that additional work is required before it is published, we would like to draw your attention to changes in our revision policy that we have made in response to COVID-19 (https://elifesciences.org/articles/57162). First, because many researchers have temporarily lost access to the labs, we will give authors as much time as they need to submit revised manuscripts. We are also offering, if you choose, to post the manuscript to bioRxiv (if it is not already there) along with this decision letter and a formal designation that the manuscript is "in revision at *eLife*". Please let us know if you would like to pursue this option. (If your work is more suitable for medRxiv, you will need to post the preprint yourself, as the mechanisms for us to do so are still in development.)

In this work, the authors demonstrate a CNN-based algorithm for the selection of embryos at 113 hours post-insemination. Although a number of machine learning algorithms for embryo classification have been reported, the value of this work is that it demonstrated improved selection of embryos at 113 hpi.

There were, however, several important issues raised during review and discussion that need to be addressed in a revised manuscripts.

1) There are several types of classification and inference performed in the study, but it was often hard to follow the order and logic of each classification. It would help immensely to have a figure that outlines the overall logic of the classification process – what the objective of the classification/inference is and how it fits into the overall embryo selection process. As it is, Figure 1 doesn't really accomplish this, and it made it difficult to follow the manuscript in places.

2) In the Introduction, discussion on the importance of evaluating embryos based on transfer outcomes needs to be strengthened. In the current version, the authors discuss on the lack of time-lapse imaging systems in fertility clinics to justify for the novelty of the work. However, to me, the major contribution of this work is the application of CNN for evaluating embryo quality at 113 hpi.

3) There was not a clear justification for why 113 hpi was used or what would be expected if other times were used.

4) It was not fully clear to the reviewers how the ground truth was established and whether there is any data on how good this ground truth is. Similarly it was not entirely clear whether any of the human annotation was used in the predictions as opposed to in training. We assume not as the authors describe this as a fully automated system, but this needs to be clarified.

5) The description of the images used should be strengthened. As it is the central data for the paper is not adequately described, with respect to image quality, completeness, etc…

6) The authors distinguish between 5 classes: 1-2 are non-blastocyst and 3-5 are blastocyst classes. They claim 90% accuracy in separating these. This should be a trivial task unless many embryos are between Morula and Early Blast (classes 2 and 3), but they don't provide a table to show the class distribution. A confusion matrix would be informative.

7) For 113 hpi blastocyst selection, the authors only report the accuracy values for both SET and DET. However, it is important to know whether the algorithm marks an embryo as blastocyst when it really is not (i.e. false positive) or as not-blastocyst when it really is (i.e. false negative). So, can the authors add a confusion matrix to show the data for all 4 cases?

8) The authors use a genetic algorithm (GA) to rank embryos. While the supplement provides a clearer description of what they do, the "genetic algorithm" part of Figure 1A is a bit misleading. In general, the Genetic Algorithm is not described well. To calculate the ranking, they multiply the probability scores from the CNN with a 5x1 weight matrix. The GA is used to optimize the weight matrix, and is not used during inference. It would be interesting to see the trained weights of this matrix – it would explain how much each class contributes to ranking.

9) It is unclear why the 5 classes were reduced to 2 for some of the analyses.

10) There is inconsistent naming of models. Early on they use a CNN, they later combine the CNN and genetic algorithm (subsection “Evaluation of embryo selection based on embryo quality”, first paragraph) and after that begin using the term system. It's not clear whether the latter two are the same or different.

11) They have an undefined term HQB which makes it hard to understand how experiments in the last paragraph of the subsection “Evaluation of embryo selection based on embryo quality” different from each other.

12) In the subsection "Evaluation of selection using implantation outcomes", they do not provide any rationale for using only fresh embryos. They simply state that they do, and alter combine the dataset with frozen embryos.

13) They aren't clear about what the implantation potential is, possibly the probability from the softmax of the CNN.

14) 5 embryos originally selected by the model had known outcome in a subsequent frozen transfer, and 4 of them led to successful implantation. This is nice, but what about the remaining 49 with an unknown outcome. I don't think any conclusion can be drawn based on the 5 with known outcomes. More data like this is needed.

15) In the Materials and methods section, the authors write that 3469 recorded videos of embryos were collected. How have the images at fixed time-points been obtained from these videos? Have you processed the images in any manner? The authors should describe more in detail how the data (i.e. images) were processed prior to feeding them to CNN for training.

16) Some of the acronyms (HQB, 3CC) appear in the Results section without full names. Although they are written in the Materials and methods sections, considering the order of the manuscript (Results then Materials and methods as currently it is), they should appear in the Results section.

17) In the Discussion, it would be helpful if the authors commented, based on their results, on the potential advantage or limitation of using a single static image for this purpose, as opposed to several images or a video clip.

18) The authors should cite and discuss the publication by Tran et al., 2019, and compare/contrast how this current submission differs and or adds to the existing literature.

19) The sentences near the end of the subsection “Evaluation of Euploid embryos based on their implantation potential” are important but are grammatically incorrect and therefore hard to understand. I get what they are trying to say but it's just poorly worded. On this note, there are a number of grammatical errors throughout the paper.

20) Citations basically stop at the Discussion section and there are some statements that definitely need literature support.

---

## [Author Response]

[…] 1) There are several types of classification and inference performed in the study, but it was often hard to follow the order and logic of each classification. It would help immensely to have a figure that outlines the overall logic of the classification process – what the objective of the classification/inference is and how it fits into the overall embryo selection process. As it is, Figure 1 doesn't really accomplish this, and it made it difficult to follow the manuscript in places.

Thank you for your comment. We have now updated Figure 1 of the revised manuscript to reflect the work described more accurately.

2) In the Introduction, discussion on the importance of evaluating embryos based on transfer outcomes needs to be strengthened. In the current version, the authors discuss on the lack of time-lapse imaging systems in fertility clinics to justify for the novelty of the work. However, to me, the major contribution of this work is the application of CNN for evaluating embryo quality at 113 hpi.

Thank you for your comment. The major contribution of our work is the development and evaluation of deep-learning models that can help identify embryos capable of implantation and in identifying the top-quality embryos from embryo sets. We have evaluated two approaches using static 113 hpi embryos images in this work: (i) the utility of a CNN in combination with a genetic algorithm for rank-ordering embryos based on their quality and (ii) utility of a CNN for the direct evaluation and estimation of the implantation potential of an embryo based on its morphology. While our method of utilizing static images for both of the approaches greatly contributes towards improving the accessibility to AI technologies for most fertility centers, our development of neural networks for the evaluation of euploid embryos is unique. Euploid embryos are usually very high-quality embryos and thus are difficult to differentiate between, based on their implantation potential through visual inspections. However, since currently there exists no assay or technology that can help differentiate between these embryos, embryologists rely on their intuition and judgment. In this work, we have shown that the network designed to measure implantation potential based on the embryo morphology at 113 hpi, outperformed trained embryologists in identifying implanting embryos from a set of PGS euploid embryos. We have now modified the Introduction of the manuscript to further clarify the significance of the reported work.

3) There was not a clear justification for why 113 hpi was used or what would be expected if other times were used.

We have now provided a rationale for using 113 hpi embryo images in the Introduction of the revised manuscript. Other time points can also be used; however, the networks need to be trained using data from that timepoint since embryo are biological cells and the morphology changes rapidly in a matter of hours.

“Conventionally, embryo transfers are performed at the cleavage or the blastocyst stage of development. […] Therefore, in this study, we have investigated the use of a CNN pre-trained with 1.4 million ImageNet images and transfer-learned using 2440 static human embryo images recorded at a single time-point of 113 hours post insemination (hpi) for the development of neural networks that can help identify embryos capable of implantation and for identifying the top quality embryos (Figure 1).”

“While this work focuses primarily on the utility of deep-learning algorithm for embryo evaluations at 113 hpi, it is also possible to develop similar networks for embryo evaluations at different timepoints, provided that sufficient data with matched outcomes/annotations are available. We have evaluated a similar network for use with cleavage-stage embryos (70 hpi) and showed that deep-learning approaches can outperform trained embryologists in certain tasks such as embryo selection (Thirumalaraju et al., 2019).”

4) It was not fully clear to the reviewers how the ground truth was established and whether there is any data on how good this ground truth is. Similarly it was not entirely clear whether any of the human annotation was used in the predictions as opposed to in training. We assume not as the authors describe this as a fully automated system, but this needs to be clarified.

The ground truth of embryo quality grading was established through manual grading annotations provided by embryologists. The ground truth in the implantation study is based on clinical implantation outcomes that were established through ultrasound inspection, 6 weeks after embryo transfer. No user input was used/needed by the algorithm in deciding the rank order and the implantation potential of the embryos during the test phase. We have now provided additional clarifications in the Materials and methods section of the revised manuscript.

“All embryos used in the study were annotated using images from the fixed time-points (113 hpi) by senior-level embryologists with a minimum of 5 years of human IVF training. Annotations for embryo implantation were assigned based on clinical outcomes.”

“Implantation signifies the attachment of a blastocyst into the endometrium. The status of implantation was clinically verified by ultrasound ~6 weeks after embryo transfer.”

“No human interaction was required/performed once the images were provided to the system during testing, as the entire process was fully automated.”

5) The description of the images used should be strengthened. As it is the central data for the paper is not adequately described, with respect to image quality, completeness, etc…

We have now provided additional information on the image collection and pre-processing prior to use with CNN in the Materials and methods section of the revised manuscript.

“The videos were collected using a commercial time-lapse imaging system (Vitrolife Embryoscope). […] Only images of embryos that were completely non-discernable were removed from the study as part of the data cleaning procedure.”

6) The authors distinguish between 5 classes: 1-2 are non-blastocyst and 3-5 are blastocyst classes. They claim 90% accuracy in separating these. This should be a trivial task unless many embryos are between Morula and Early Blast (classes 2 and 3), but they don't provide a table to show the class distribution. A confusion matrix would be informative.

We have now provided a confusion matrix (Figure 2—figure supplement 1) in the revised manuscript.

7) For 113 hpi blastocyst selection, the authors only report the accuracy values for both SET and DET. However, it is important to know whether the algorithm marks an embryo as blastocyst when it really is not (i.e. false positive) or as not-blastocyst when it really is (i.e. false negative). So, can the authors add a confusion matrix to show the data for all 4 cases?

The SET and DET selections are performed after rank ordering with the scores obtained from the combination of the CNN SoftMax probabilities and the genetic algorithm weights. Essentially, the use of genetic algorithm (GA) helps in selecting only embryos of the highest quality and confidence by generating a rank order and thereby aids in minimizing the error of the CNN system. Therefore, a confusion matrix is not suitable to understand the overall algorithm. However, a confusion matrix can help better understand the CNN classifier (without the genetic algorithm). We have now provided the confusion matrix of the CNN in Figure 2—figure supplement 1.

8) The authors use a genetic algorithm (GA) to rank embryos. While the supplement provides a clearer description of what they do, the "genetic algorithm" part of Figure 1A is a bit misleading. In general, the Genetic Algorithm is not described well. To calculate the ranking, they multiply the probability scores from the CNN with a 5x1 weight matrix. The GA is used to optimize the weight matrix, and is not used during inference. It would be interesting to see the trained weights of this matrix – it would explain how much each class contributes to ranking.

We have now provided the final weights generated by the genetic algorithm in the Results section. The genetic algorithm tries to heavily de-emphasize class 1 (arrested/degenerate) embryos. Interestingly, it also tries to avoid selecting embryos that are early stage blastocysts. Unsurprisingly, It prioritizes features that are associated with the highest quality embryos during selection.

“…once we established that the network was capable of categorizing embryos based on their morphologies with high accuracy, we utilized the network defined probability values of the embryos, belonging to each of the 5 training classes, with a genetic algorithm to rank order the embryos for transfer. […] The final weights utilized in this study were -10.01226347, -3.63697951, -3.32090987, 2.15367795, and 2.8715555 for classes 1 through 5, respectively.”

9) It is unclear why the 5 classes were reduced to 2 for some of the analyses.

The 2 deep learning models used two classification systems. We have now clarified them in more detail in the Materials and methods section of the revised manuscript.

“The two networks in this study used two categorization systems. The network focused on the rank ordering of embryos used a hierarchical categorization system. […] Networks that were focused on estimating an embryo’s implantation potential used a two-class training and inference system- positive for implantation and negative for implantation.”

10) There is inconsistent naming of models. Early on they use a CNN, they later combine the CNN and genetic algorithm (subsection “Evaluation of embryo selection based on embryo quality”, first paragraph) and after that begin using the term system. It's not clear whether the latter two are the same or different.

In this study, we have evaluated two deep learning-based approaches for identifying embryos capable of implantation. One version of the neural networks was designed to classify, and rank order embryos based on their morphological quality and the other version was designed to classify embryos based on the implantation potential. The two networks share a common Xception architecture, but the classification layers are specific to each task. We have now expanded the Introduction section of the revised manuscript to improve clarity on the different approaches evaluated in this study.

“Therefore, in this study, we have investigated the use of a CNN pre-trained with 1.4 million ImageNet images and transfer-learned using 2440 static human embryo images recorded at a single time-point of 113 hours post insemination (hpi) for the development of neural networks that can help identify embryos capable of implantation and for identifying the top quality embryos (Figure 1). […] The CNN was trained and evaluated in identifying euploid embryos capable of implantation and the performance was compared against those of 15 embryologists from 5 different fertility centers across the United States of America.”

11) They have an undefined term HQB which makes it hard to understand how experiments in the last paragraph of the subsection “Evaluation of embryo selection based on embryo quality” different from each other.

HQB stands for high-quality blastocysts, which is now defined in the Results section of the revised manuscript. We have also provided additional clarifications to improve the clarity on rationale and design of the experiments that were performed.

“According to the American Society for Reproductive Medicine guidelines on the limits to the number of embryo transfer, 1 embryo is transferred for high prognosis patients with <37 years of age and 2 or more embryos are transferred for patients with >37 years of age as well as younger patients with low prognosis (Guidance on the limits to the number of embryos to transfer: a committee opinion, 2017). […] High-quality blastocysts are defined as embryos that met the freezing criteria (>3CC blastocyst grade; see Materials and methods) of the Massachusetts General Hospital (MGH) fertility clinic.”

12) In the subsection "Evaluation of selection using implantation outcomes", they do not provide any rationale for using only fresh embryos. They simply state that they do, and alter combine the dataset with frozen embryos.

We have now included our rationale on why fresh cycles were used in evaluating the networks in the revised manuscript.

“Typically, in a clinical IVF cycle, the top-quality embryo is selected from the cohort of available embryos and is transferred to the patient. […] The test dataset was retrospectively collected based on pre-defined selection criteria and evaluations of transfer outcomes were performed using fresh embryo transfer cycles.”

“A limitation of a retrospective study is that not all embryos are transferred. Implantation outcomes of all embryos selected by the algorithm cannot be evaluated. Therefore, although the dataset was prepared not taking into consideration the availability of subsequent frozen cycle transfers, we investigated all patients from the test set who had subsequent embryo transfers using frozen embryos from the test set.”

13) They aren't clear about what the implantation potential is, possibly the probability from the softmax of the CNN.

We have now explicitly stated in the Results section that implantation potential is the SoftMax output of the CNN.

“The network was trained as a binary classifier and the SoftMax probability values outputted by the network was used as the embryo’s implantation potential.”

14) 5 embryos originally selected by the model had known outcome in a subsequent frozen transfer, and 4 of them led to successful implantation. This is nice, but what about the remaining 49 with an unknown outcome. I don't think any conclusion can be drawn based on the 5 with known outcomes. More data like this is needed.

The frozen cycle transfers evaluated in this study was not one of the primary objectives. Typically, in a clinical IVF cycle, the top-quality embryo is selected from the cohort of available embryos and is transferred to the patient. Embryos, which are similarly of a high-quality, are often frozen based on the freezing criteria used by the fertility center, for transfers in subsequent procedures for the same patient if needed. Frozen cycle transfers are not performed for all patients. Therefore, the retrospective test set of patient embryo cohorts (n=97) was selected, and evaluations of transfer outcomes were performed using fresh embryo transfer cycles. A limitation of such a retrospective study is that not all embryos are transferred. Therefore, although the dataset was prepared not taking into consideration the availability of subsequent frozen cycle transfers, we investigated with the fertility center if the patients of the test set had any subsequent embryo transfers using the frozen embryos from the test set and reported our findings.

15) In the Materials and methods section, the authors write that 3469 recorded videos of embryos were collected. How have the images at fixed time-points been obtained from these videos? Have you processed the images in any manner? The authors should describe more in detail how the data (i.e. images) were processed prior to feeding them to CNN for training.

We have now provided more information on the pre-processing that was performed on the input images prior to feeding them to the CNN for training and testing in the Materials and methods section of the revised manuscript. Briefly, the images were only cropped to remove timestamps and identifiers. No additional image pre-processing was performed.

“The imaging system used a Leica 20x objective that collected images at 10 min intervals under illumination from a single 635 nm LED. […] Only images of embryos that were completely non-discernable were removed from the study as part of the data cleaning procedure.”

16) Some of the acronyms (HQB, 3CC) appear in the Results section without full names. Although they are written in the Materials and methods sections, considering the order of the manuscript (Results then Materials and methods as currently it is), they should appear in the Results section.

We have now provided the full names of the acronyms at their first instance in the revised manuscript.

17) In the Discussion, it would be helpful if the authors commented, based on their results, on the potential advantage or limitation of using a single static image for this purpose, as opposed to several images or a video clip.

We have now discussed the benefits and challenges of using several images/video clips for embryo assessment in the revised manuscript.

“The network’s learning can be compounded with data from additional timepoints, morphokinetics, and patient and cycle-specific information for more personalized IVF predictions and outcomes…”

“A major hurdle for the development of networks capable of analyzing multi-timepoint images and with additional patient-specific information is the limited availability of diversified data with known clinical outcomes. […] The need for data scales with the complexity of the task and the number of variables introduced.”

18) The authors should cite and discuss the publication by Tran et al., 2019, and compare/contrast how this current submission differs and or adds to the existing literature.

We had cited the work by Tran et al. in the first version of the manuscript. However, in the revised manuscript we have now added a discussion around their work. Briefly, we mention that the work by Tran et al. was flawed due to improper study design and thus is of limited value. Furthermore, the focus of Tran et al., is in utilizing only time-lapse videos, thereby restricting their approach to expensive, bulky, and rarely available instruments preventing its utility for most fertility centers in the US. In contrast, our approach focuses on single timepoint images enabling potential access to critical future technology. Additionally, our work has shown for the first time, the ability of a neural network to identify embryos capable of implantation amongst similar quality euploid embryos.

Discussion:

“Recently, Tran et al. studied the use of a deep-learning model (IVY) that can analyze whole time-lapse videos instead of specific time points for fetal heartbeat prediction (Tran et al., 2019). […] A major hurdle for the development of networks capable of analyzing multi-timepoint images and with additional patient-specific information is the limited availability of diversified data with known clinical outcomes.”

“Clinically, besides morphological features, various other important metrics and parameters are considered by embryologists at the time of decision making such as taking into account the ploidy status of the transferable embryos. […] The CNN-based approach, through direct estimations of implantation potential from 113 hpi embryo morphology, outperformed trained embryologists in identifying implanting embryos from a set of PGS euploid embryos.”

19) The sentences near the end of the subsection “Evaluation of Euploid embryos based on their implantation potential” are important but are grammatically incorrect and therefore hard to understand. I get what they are trying to say but it's just poorly worded. On this note, there are a number of grammatical errors throughout the paper.

Thank you for your comment. We have now rephrased these sentences in the revised manuscript and have also added a supplementary figure for clarity. We have also corrected all typographical errors throughout the revised manuscript.

“Approximately 91% of euploid embryos with implantation potential scores of 0.80 or higher, and nearly 81% of embryos with implantation potential scores above 0.66 successfully implanted when transferred (Figure 3—figure supplement 1). […]These results suggest that the network’s implantation scores agree well with transfer outcomes even in high-quality euploid embryos.”

20) Citations basically stop at the Discussion section and there are some statements that definitely need literature support.

We have now added additional references in the Discussion section where it was needed.